# The Use of Social Media for Preconception Information and Pregnancy Planning among Young Women

**DOI:** 10.3390/jcm10091892

**Published:** 2021-04-27

**Authors:** Helen Skouteris, Melissa Savaglio

**Affiliations:** 1Health and Social Care Unit, School of Public Health and Preventive Medicine, Monash University, Melbourne, VIC 3004, Australia; melissa.savaglio@monash.edu; 2Warwick Business School, The University of Warwick, Coventry CV4 7AL, UK

**Keywords:** pregnancy, preconception, social media, health information

## Abstract

Young women of reproductive age (18–25 years) often seek and engage with health-related information via social media. However, the extent to which this population group seek preconception and pregnancy-related information has not been explored. Therefore, this study aimed to: (1) examine the proportion, type, and frequency of social media use to seek general health, preconception, and pregnancy-related information or advice among young women; and (2) explore the relationship between age, education status, relationship status, and planning a pregnancy on social media use for preconception and pregnancy-related health information. Ninety-one Australian women aged 18–25 years completed an online survey about their patterns and preferences of social media use for this information. Forty percent of women used social media for general health information (most often daily), 32% for preconception health advice (most commonly weekly), and 20% for pregnancy-related information (most often weekly), with Facebook the most frequently used platform. Planning a pregnancy within the next 5 years and younger age were associated with greater likelihood of using social media for such information. It is necessary that social media platforms are leveraged to disseminate preconception and pregnancy planning health advice, support, and education to provide better health promotion and preventive care to young women.

## 1. Introduction

Women during preconception and pregnancy planning periods often engage in increased information seeking, decision-making behaviours, and lifestyle changes to maximise individual and intergenerational health [1]. With advances in social media and technology in recent years, young women of reproductive age (i.e., 18–25 years old) are turning to social media for health-related information, such as lifestyle advice, that was usually exclusively sought directly from health care providers, family, peers, or printed media [2]. Specifically, this age group of young women have the highest rate of social media use and engagement [3]. In this paper, social media refers to internet-based social network platforms and media sharing sites or mobile phone applications (apps) that allow individuals to communicate, share content, and collaborate, such as Facebook, Instagram, and Twitter [4,5].

Social media has become one of the easiest, convenient, readily accessible, and time-saving ways for young women of reproductive age to learn about health during preconception and pregnancy planning periods [6,7]. In addition to numerous specific pages, groups, and accounts dedicated to preconception and pregnancy care on social media platforms, such as Facebook and Instagram, there are currently more pregnancy-related apps than any other health or fitness app [8]. Health professionals are also increasingly using social media themselves to communicate health messages and post health information in more accessible forms, such as videos and images, which can be retrieved from any place at any time [9]. Key benefits of social media use for health information, from the perspective of end-users themselves, include increased accessibility to relevant and tailored advice, greater interactivity, self-monitoring, convenient, little to no cost, and more timely answers [2,7,10,11]. Social media also allows its users to share health information with others, including their health care experiences, symptoms, and knowledge [2,12], which can facilitate social support, connection, and a sense of ‘community’ for women of young reproductive age [5].

However, social media use for seeking health information has its limitations. The content shared on social media can at times be of poor quality, conflicting, inaccurate, misleading, or not credible [13,14]. Misinformation derived from apps has been shown to increase worry, stress, and false reassurance, which can translate to poorer health outcomes [13]. Therefore, health professionals require a greater presence on social media to connect women with credible information, reduce the amount of misinformation, and provide accessible services, especially for those with poor access or social isolation [14]. Social media still faces challenges in user adoption and participation regarding health information as it requires end users to actively participate, react to, interact with, and share online information [15]. Specifically, privacy risk or concerns due to the sensitive nature of some health-related content may deter users from participating and interacting with information on social media [16].

Few studies have examined the intentions of patients, health seekers, or lay users (i.e., non-health care professionals) to seek and share pregnancy and preconception health information on social media [5]. Yet, women who are pregnant or planning for pregnancy have been identified as a population group who are highly motivated to seek out information online, particularly to alleviate uncertainty about health-related decisions, seek reassurance, and desire to share their experiences with others [17,18]. Studies have found that women prefer information that is immediate, regular, detailed, practical, reassuring, and customised [18,19,20]. Lifestyle management programs targeting women who are pregnant that advertise health advice messaging through social media and provide question and answer support have been found to improve pregnancy care, effectively reduce gestational weight gain, increase activity levels, and lower perinatal depressive symptoms [21,22].

Targeting health promotion messages to young women (aged 18–25 years old) has been shown to yield sustainable behaviour changes (e.g., in areas of substance use, physical activity, diet, tanning etc.) [3,23]. This preventive approach can translate to better individual health outcomes over their life course (i.e., prevention of disease), as well as improved intergenerational health for their children [1]. However, to the authors’ knowledge, no study to date has examined the proportion, type, and frequency of social media use for preconception and pregnancy information among young women of reproductive age (18–25 years old). Furthermore, the limited existing literature predominantly focuses on pregnancy care, often neglecting preconception. This is a clear gap as these young women tend to engage in unhealthy lifestyle behaviours that may adversely impact their preconception health, pregnancy planning, or fertility, such as binge drinking, poor diet (i.e., this age group is least likely to meet Australian Dietary guidelines for fruit and vegetable consumption), low rates of contraception use, and high rates of sexually transmitted diseases [24,25,26]. Furthermore, these women have high rates of unplanned pregnancies that are often associated with delayed initiation of antenatal care, poor maternal health, and adverse birth outcomes [27,28].

It is necessary to explore the social media use of young reproductive age women and to also understand the characteristics of those who are more likely to use it so that health promotion information can be appropriately targeted and disseminated accordingly to yield sustainable change. Therefore, this study aimed to: (1) examine the proportion, type and frequency of social media use to seek general health, preconception, and pregnancy-related health information or advice among young adult women of reproductive age; and (2) explore the relationship between key demographic characteristics (age, education status, relationship status, and planning a pregnancy) on social media use for preconception and pregnancy-related health information.

## 2. Materials and Methods

### 2.1. Design and Ethics

This study utilised a cross-sectional survey design. Ethics approval was obtained from Monash University Human Research Ethics Committee (project number 23619).

### 2.2. Participants

Women aged 18–25 years who lived in Australia and could read and write in English were eligible to participate in this study. No other exclusion criteria applied. Women were invited to participate in an online survey via advertisements on social media (i.e., Facebook and Twitter)—including specific pregnancy, fertility, and mum groups on Facebook— and Australian online pregnancy forums, and via links posted on various university student online web forums throughout. A total of 93 women responded and completed the survey.

### 2.3. Survey

A 5–10 min online Qualtrics survey was developed by the research team to explore women’s social media use regarding preconception and pregnancy-related information (please see Appendix A). The survey asked women to report key demographic characteristics and reflect on their social media use when seeking information about (1) general health; (2) preconception health; and (3) pregnancy health. Key demographic characteristics included age, level of education, relationship status, employment status, annual income, whether they had children, and whether they were planning a pregnancy in the near future. The survey also asked women to identify the following: (1) extent of knowledge about pregnancy from previous or current education; (2) whether they use social media to seek information related to general health, preconception, and pregnancy; (3) the types of social media platforms (i.e., Facebook, Instagram, Twitter, Snapchat, Tumblr, and/or other) and their frequency of use to seek information related to general health, preconception, and pregnancy; (4) which types of social media platforms they would seek to gain this information in the future; and (5) whether women share preconception and/or pregnancy information from social media with others.

### 2.4. Procedure

Women who expressed interest in the study and responded to the advertisement were provided a web address link to complete the 5–10 min Qualtrics online survey. Women could complete the survey on their phone, tablet, or computer at any time that was convenient for them. Completion of the survey implied consent, and all responses remained anonymous. On average, it took approximately 6 min (SD = 3.28 min) for the women to complete the online survey. Participants did not receive any incentive or compensation for their participation.

### 2.5. Data Analysis

Analyses were performed using SPSS Statistics Version 27.0 (IBM, Sydney, New South Wales, Australia). The data were coded, cleaned, and managed to account for any missing data; there were two participants with substantial missing data, so their data were excluded from analyses, resulting in a final sample of 91 women. Data were screened to ensure that they conformed to assumptions for regression analyses. Descriptive statistical analyses were conducted for demographic variables and social media use of women. Standard binomial logistic regression analyses were conducted to determine if age, education status, relationship status, and planning a pregnancy predicted women’s social media use for seeking preconception and pregnancy-related health information. Statistical significance was set at a = 0.05. All related assumptions were met. The final sample size for this exploratory study (*n* = 91) gave sufficient power based on an alpha level set at 0.05, a moderate effect size of 0.15 (based on similar exploratory studies with this population group) [29], and 80% power to identify an association between predictor variables and social media use (G*Power, version 3.1.9.6, Heinrich Heine University, Dusseldorf, Germany). Furthermore, according to Green’s (1991) formula [30] of 50 + 8k, where k is the number of predictors, our sample size exceeded the minimum number required for four predictors (*n* = 82). Sample size calculations were conducted prior to recruitment.

## 3. Results

### 3.1. Demographic Characteristics

A total of 91 women of young reproductive age from 18–25 years completed the survey. Key demographic characteristics are presented in Table 1. Women were 21 years of age on average, were currently or had finished completing a bachelor’s degree, and were predominantly working in casual employment. Just over half of the women were not in a relationship. The majority of women (93%) did not have any children, and 73% were not planning a pregnancy in the near future (i.e., within the next 5 years). The women provided the following explanations for why they were not planning for a pregnancy in the near future, which included: wanting to finish studying (33%), too young (28%), work/career commitments or aspirations (16%), do not want children or to be pregnant (12%), want to travel first (10%), become more financially stable (8%), no partner (7%), uncertain about wanting children (4%), health concerns (1%), and intend to adopt (1%). Women could choose more than one reason.

### 3.2. Social Media Use for General Health Information

The various types of social media platforms that the women reported using to gain general, preconception, and pregnancy-related health information are presented in Table 2. Forty percent of women (*n* = 36) reported using a social media app/platform to gain information about their general overall health. The most common types of social media platforms reported by these women included Facebook (89%) and Instagram (89%), and approximately half used Snapchat (53%). Seven women (19%) each endorsed an ‘other’ type of social media, such as Tik Tok and period-tracking apps (i.e., Flo and Clue). The frequency of social media use for each type of information is presented in Table 3. The women who used social media to seek general health information most commonly reported daily use.

### 3.3. Social Media Use for Preconception Information

Thirty-two percent of women (*n* = 29) reported using social media platforms to gain information about preconception health. The majority of these women endorsed Facebook (62%) as their preferred source of information, and half reported using Instagram (54%). Further, 21% of these women nominated Snapchat as a source of such information, and three women reported using ‘other’ apps, including Flo Period Tracker (*n* = 1) and TikTok (*n* = 2). The frequency of social media use for preconception health information was most often weekly. Further, only 13% reported sharing preconception health information from social media pages with their family and peers, by sharing Facebook posts, Instagram stories, commenting on posts, retweeting, or privately messaging social media posts to friends. Finally, 68% of women reported that they would use social media to seek preconception health information in the future, with Facebook the most preferred choice for future use, followed by Instagram.

The results of standard binary logistic regression to ascertain the effects of age, education status, relationship status, and planning a pregnancy on the likelihood that women use social media for preconception-related health information are presented in Table 4. The overall model was statistically significant, χ^2^(9) = 21.812, *p* = 0.019. The model explained 30% (Nagelkerke R^2^) of the variance in social media use and correctly classified 73% of cases. Planning a pregnancy was the strongest predictor, with those who expressed near-future intention four times more likely to use social media for preconception-related information than those who were not planning a pregnancy in the next 5 years (*p* = 0.02). However, age, relationship, and education status were not statistically significant predictors of social media use for preconception information.

### 3.4. Social Media Use for Pregnancy Information

The majority of women (78%) reported having adequate information about pregnancy from their past or current education, particularly from high school (45%) and university (34%) education. Other women reported receiving information about pregnancy from family and friends (15%), through their own personal experience of pregnancy (3%), and by conducting maternal health research (3%). Twenty percent of women (*n* = 18) reported using a social media platform to gain information about pregnancy. The most used type of social media for pregnancy health-related information was Facebook (67%), followed by Instagram (50%). Three women reported using an ‘other’ social media platform, including Pregnancy+ (*n* = 1), and TikTok (*n* = 2). Frequency of social media use for pregnancy information was most often weekly. Further, 12% of women reported sharing pregnancy health-related information gained from social media platforms with their family and peers, by sharing Facebook posts, Instagram stories, commenting on posts, retweeting, or privately messaging social media posts to friends. In terms of future use, 60% of women reported that they would use social media for pregnancy information, with Facebook the most preferred option.

The results of standard binary logistic regression to ascertain the effects of age, education status, relationship status, and planning a pregnancy on the likelihood that women use social media for pregnancy-related health information are presented in Table 5. The overall model was statistically significant, χ^2^(9) = 18.741, *p* = 0.027. The model explained 31% (Nagelkerke R^2^) of the variance in social media use and correctly classified 82% of cases. Similar to preconception information, planning a pregnancy was the strongest predictor, with those who expressed near-future intention 6.6 times more likely to use social media for pregnancy-related information than those who were not planning a pregnancy in the next 5 years (*p* = 0.008). Increasing age was associated with decreased likelihood of using social media for pregnancy-related information (*p* = 0.03). Relationship and education status were not statistically significant predictors of social media use.

## 4. Discussion

Targeting young women (aged 18–25 years old) in health promotion messages can yield sustainable change in lifestyle behaviours, which can translate to better individual and intergenerational health outcomes [1,3]. Social media platforms present an avenue to leverage preconception and pregnancy health information among young women, yet this has not been explored to date. Therefore, this study aimed to: (1) examine the proportion, type, and frequency of social media use to seek general health, preconception, and pregnancy-related health information or advice among young women of reproductive age; and (2) explore the relationship between key demographic characteristics and social media use for such information. There were 40% of women who reported using social media for general health information, 32% for preconception health advice, and 20% for pregnancy-related information. These proportions are similar to research demonstrating that approximately 25% of women use social media to learn about health care topics [19]. However, use of established social media platforms for preconception and pregnancy health information remains significantly lower than specific mobile apps, such as those related to pregnancy (i.e., tracking apps), which are routinely used by approximately 55–75% of pregnant women [17,18]. In terms of frequency of use, social media was utilised daily for general health information, and weekly for information relating to pregnancy or preconception health. This is a novel finding that provides insight into the patterns of information-seeking behaviours among young women of reproductive age.

Facebook was the most common type of social media platform used by the women, followed closely by Instagram. Facebook was also consistently the most preferred option for future social media use for preconception and pregnancy health information. Facebook is currently the most popular social media platform, and the literature has consistently focused on Facebook as the primary health information sharing platform. Specifically, engagement in a Facebook group page for expectant mothers has been associated with expressive information sharing, social interaction, and information seeking [11]. Further, Facebook posts have shown to provide more general pregnancy-related information or opportunities for personal sharing, while Instagram posts have more emotional support posts intended to make pregnancy relatable, which are actually viewed more favourably [20]. Other social media platforms (i.e., Snapchat, Tik Tok) were also acknowledged by women, yet these sites have been overlooked in the literature to date when it comes to preconception and pregnancy information for reproductive age women, particularly those aged under 25 years. Whilst it is necessary to target platforms that are most used, it is also recommended that efforts are directed towards sharing preconception and pregnancy-related information across various social media platforms to ensure that information is wide-reaching among the general population.

The current findings also provide some insights about the characteristics of young women who are more likely to use social media to gain preconception and pregnancy-related information. Primarily, those who were planning a pregnancy in the next 5 years were significantly more likely to report seeking preconception and pregnancy-related health information via social media. This suggests that social media posts, groups, and accounts may need to be targeted and tailored to reach those specific groups of women and who are actively engaging in information-seeking behaviours. Further, use of social media was less likely with increasing age, suggesting that older women of reproductive age may seek alternative sources for pregnancy-related information (i.e., health professionals, peers, journal articles). Indeed, young adults use the Internet to seek health information more than any other source of information [9]. However, in relation to preconception health, there are numerous universal and persistent barriers to preconception care, particularly for young women, such as the lack of awareness of importance of preconception care, uncertainty around where to access such information, high rates of unplanned pregnancies, and limited resources [27,31]. Therefore, it is necessary that health professionals utilise and leverage social media as a possible solution to address this gap and provide equitable health care to young women of reproductive age during the preconception period.

This study found that only 12–13% of young women shared preconception and/or pregnancy-related health information found on social media with others. This finding is much lower than the 40–45% sharing rate found in previous studies for pregnancy-related information [7,22]. Information-seeking and sharing of health care knowledge, experiences and symptoms are important aspects of social media, which can bond those who have similar concerns [11]. The lower rate of sharing information found in the current study may be due to the demographic characteristics of the sample, with the majority of participants likely to not be part of specific online support or self-help groups related to preconception or pregnancy, where such information is primarily routinely shared [22]. These groups provide individualised informational, social, and emotional support, particularly for pregnant women [2]. Finally, the finding that approximately two-thirds of young women would intend to use social media for preconception (68%) or pregnancy (60%) health information in the future suggests that this mode of communication and information sharing should be leveraged to improve accessibility to, and quality of, health care during these life stages.

The current findings must be considered in light of some key limitations. First, given the limited literature and lack of understanding around patterns of social media use for young women, we felt that it was necessary to focus on this cohort of women aged 18–25 years, particularly from an early prevention and health promotion approach. However, it must be acknowledged that this sample is not fully representative of the broader population of women of reproductive age (which can extend up to 49 years of age). Further, participants in this study were more likely to be engaged in employment (85%) and earn a higher income than females in the general Australian population, of which 70% of females aged 20–24 years are employed [32]. Therefore, the results may not be accurately generalisable to all women who use social media for preconception and pregnancy health information across Australia; conclusions based on these findings must be considered with caution. Future research is required to validate these findings among women of various demographics (i.e., women of different cultures, socio economic status, geographical region, and older ages) in other contexts. Particularly, the methodology of an online survey may have introduced bias relating to the characteristics of the sample. Women with greater income, higher socioeconomic status, higher education, internet access, and those with general preferences to seek information online, may have been more likely to access the online survey for this study. Indeed, research methodologies and interventions using digital platforms often risk being exclusionary to key population groups [33]. The use of such platforms must be reconsidered to ensure equitable access and engagement. Finally, this study did not examine women’s experiences and evaluation of using social media for such information. There is little research in terms of the acceptability, helpfulness, usefulness, and type of pregnancy and preconception health information sought via various social media platforms. Addressing these gaps should be a priority for future research so that information that is disseminated via social media can be accurate, appropriate, accessible, high quality, and responsive to young women’s needs to improve their health care.

This study has provided new insights about the current landscape of social networking platforms, in terms of how young reproductive-aged women interact and seek advice on social media regarding preconception and pregnancy health. This increased knowledge may help key stakeholders, such as health professionals, researchers, health care organisations, public health groups, and policy-makers in leveraging social media, particularly Facebook and Instagram, to provide preconception and/or pregnancy-related health care, information, education, and support. Specifically, it is recommended that all health care organisations and public health groups harness the potential social media by using Facebook and Instagram (preferably both platforms) to communicate credible, interactive, and clear health promotion messages to young women of reproductive age. Specific strategies may include: targeted health care social media marketing to raise awareness about public health concerns relevant to preconception and pregnancy planning for this age group (i.e., unplanned pregnancies); initiating, monitoring, and answering questions in online discussion groups to facilitate a sense of ‘community’ and provide support; regularly posting health information in accessible forms, such as video and images; and enhancing the visibility of their organisation through promotion of their service. Indeed, the current findings around the proportion, type, and frequency of social media use could help inform the design and dissemination of information via social media. Such information that appropriately targets the health-seeking behaviours of specific groups of women of reproductive age could improve health outcomes and facilitate equitable health care. It is necessary to improve the provision of relevant, accurate, and evidence-based preconception and pregnancy health advice with social support to more effectively reach young women in this increasingly technology-driven society. This may have the potential to yield sustainable change and improve their long-term health outcomes.

## 5. Conclusions

Social media platforms present key opportunities to provide evidence-based health information, such as lifestyle advice, to pregnant and preconception women. The current findings highlight that a significant portion of young women of reproductive age use social media to seek information related to preconception and pregnancy health, particularly those who are planning for a pregnancy. It is therefore crucial that key stakeholders such as health professionals, health care organisations, public health groups, and researchers utilise social media platforms to disseminate quality health advice, support, and education to better provide health care equity to all women during preconception and pregnancy planning periods.

## Figures and Tables

**Table 1 jcm-10-01892-t001:** Demographic characteristics.

Demographic Variable	Frequency (%) (*n* = 91)
Mean age (SD)	21.18 (1.99)
**Educational level**	
Did not finish secondary school	2 (2)
Year 12 or equivalent	22 (24)
Certificate level	8 (9)
Advanced diploma	1 (1)
Bachelor’s degree	48 (53)
Higher university degree (Masters, PhD)	10 (11)
**Employment status**	
Unemployed	14 (15)
Casual employment	41 (45)
Part-time employment	18 (20)
Full-time employment	18 (20)
**Annual income**	
$1–$6239 annually	10 (11)
$6240–$15,999 annually	22 (24)
$16,000–$25,999 annually	13 (14)
$26,000–$36,999 annually	13 (14)
$37,000–$51,999 annually	14 (15)
$52,000 + annually	8 (9)
No income	8 (9)
Did not answer	3 (3)
**Relationship status**	
Single	50 (55)
Partnered	40 (44)
Married	1 (1)
**Have children**	
No	85 (93)
Yes	6 (7)
**Panning a pregnancy**	
No	66 (73)
Yes, within the next 6–12 months	0 (0)
Yes, within the next 1–5 years ^1^	25 (27)

^1^ The range for pregnancy planning was determined by women aged 18–25 years. SD-standard deviation.

**Table 2 jcm-10-01892-t002:** Types of social media that women use for general, preconception, and pregnancy health-related information.

	Type of Information *n* (%) ^1^		
Type of Social Media	General Overall Health Information	Preconception Health-Related Information	Pregnancy Health-Related Information
Facebook	32 (89)	18 (62)	12 (67)
Instagram	32 (89)	15 (52)	9 (50)
Snapchat	19 (53)	6 (21)	7 (39)
Twitter	7 (19)	4 (14)	2 (11)
Tumblr	2 (6)	3 (10)	2 (11)
Other (apps)	7 (19)	3 (10)	3 (17)
Total *n* that use social media	36 (40)	29 (32)	18 (20)

^1^ Participants could choose more than one type of social media. Percentages are based on the number of participants who reported using social media for each type of information.

**Table 3 jcm-10-01892-t003:** Frequency of social media use for each type of information.

Frequency of Use-*n* (%) ^1^				
Type of Social Media	Daily	4–6 Times a Week	2–3 Times a Week	Weekly
**Facebook**				
General health	18 (56)	0	3 (9)	11 (34)
Preconception	5 (28)	2 (11)	1 (5)	10 (56)
Pregnancy	5 (42)	0	0	7 (58)
**Instagram**				
General health	19 (59)	2 (6)	2 (6)	9 (29)
Preconception	7 (47)	0	1 (6)	7 (47)
Pregnancy	4 (44)	0	0	5 (56)
**Snapchat**				
General health	10 (53)	2 (11)	2 (11)	5 (26)
Preconception	4 (67)	1 (17)	1 (17)	0
Pregnancy	2 (29)	1 (14)	1 (14)	3 (43)
**Twitter**				
General health	3 (43)	0	1 (14)	3 (43)
Preconception	1 (25)	1 (25)	0	2 (50)
Pregnancy	0	1 (50)	0	1 (50)
**Tumblr**				
General health	1 (50)	1 (50)	0	0
Preconception	1 (33)	0	0	2 (67)
Pregnancy	0	0	0	2 (100)
**Other**				
General health	2 (29)	1 (14)	1 (14)	3 (43)
Preconception	1 (33)	0	0	2 (67)
Pregnancy	1 (33)	0	0	2 (67)

^1^ Percentages are based on the number of participants who reported using each type of social media platform for each information group.

**Table 4 jcm-10-01892-t004:** Binomial logistic regression of demographic variables associated with social media use for preconception-related information.

Predictors	B	SE B	Wald χ^2^	*p*	Ex (β)
Age	−0.206	0.172	1.434	0.231	0.814
Education status	2.532	2.932	6.728	0.242	2.203
Relationship status	0.325	0.564	0.333	0.564	0.722
Planning pregnancy	1.421	0.610	5.435	0.020 ^1^	4.142

^1^*p* < 0.05. *n* = 91. B-beta, SE-standard error, Ex-exponentiation of beta.

**Table 5 jcm-10-01892-t005:** Binomial logistic regression of demographic variables associated with social media use for pregnancy-related information.

Predictors	B	SE B	Wald χ^2^	*p*	Ex (β)
Age	−0.505	0.234	4.672	0.031 *	0.603
Education status	−0.819	1.573	0.271	0.603	0.441
Relationship status	0.037	0.638	0.003	0.954	1.038
Planning pregnancy	1.897	0.710	7.141	0.008 *	6.663

* *p* < 0.05. *n* = 91. B-beta, SE-standard error, Ex-exponentiation of beta.

## Data Availability

The data presented in this study are available on request from the corresponding author.

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
