# Peer review of "The Use of Social Media for Preconception Information and Pregnancy Planning among Young Women"

_jcm, 2021, doi:10.3390/jcm10091892_

Round 1

Reviewer 1 Report

The paper describes findings from an online survey for women in the reproductive age group (18-25). Overall an important topic as studies have shown that it is difficult to engage women who are in the preconception group, but also since most pregnancies are unplanned. 

The paper is well written, my comments are mainly related to the need to acknowledge some of the methodological limitations. The authors have not commented on the potential inequalities introduced by interventions using digital platforms, or even the bias in the methodology of an online survey itself.

the authors could comment on the issue of bias related to more high-income/ higher educational groups and women who may seek information on the internet being more likely to have accessed the online survey.

The rationale for the study has been presented well, and targeting the younger demographic of the reproductive age group was appropriate for this research questions. Please clarify if any incentive/ voucher was provided for respondents? It is surprising that the numbers have been low (were recruitment via targeted, paid social media adverts or just posting in specific groups such as pregnancy/ fertility forums, health/ nutrition discussion groups and mum's groups on Facebook?) The latter would surely pick up a motivated group of women. 

In the data analysis section, the sample size calculation is provided, but it is unclear how the estimated effect size was selected. Was the calculation done before the study? Since this was an exploratory study, perhaps it would be ok to acknowledge the aim was to understand the prevalence and patterns of social media for preconception factors use in this group, previously unexplored.

Limitations on the representativeness of the population related to the age group are mentioned in the discussion. The discussion could also provide more information on how representative the sample is for the Australian population for other factors e.g. mean income in this age group, geographical distribution. Were respondents more local to the university?

For the question on pregnancy planning - 

1-5 years is a pretty broad range for pregnancy planning, perhaps the authors could justify this selection? Most definitions consider up to 2 years for the “preconception” group.

Overall the results are interesting, however, the authors could explore further and provide some specific recommendations for future interventions/ research using digital platforms. More than healthcare professionals, the audience of this paper could be healthcare organisations and public health groups who could harness the potential of social media. Previous studies have shown that clinicians such as doctors and midwives are already missing the opportunity to discuss pregnancy intention, nutrition and other issues due to time constraints, so uncertain if they would use social media to get the message across. 

Could the questionnaire be uploaded as supplementary material please?

Author Response

Thank you for the opportunity to revise our paper for consideration for publication in JCM. We thank you for the insightful and helpful feedback regarding the manuscript; it has certainly strengthened the paper. The manuscript has been revised in accordance with your feedback. Please see the attachment for a point-by-point response to your comments.

Reviewer 2 Report

This is a descriptive study of the proportion, type, frequency of social media use for health information seeking regarding preconception and pregnancy in young women of reproductive age. 

  1. The title is misleading; it does not specify the young age range of the population and implies that there is data relevant to currently pregnant women, which is not the case.
  2. Why is the target population of the study limited to age 18-25 years? Presumably women up to age 30 and beyond also engage heavily with social media and apps and are more likely to be pregnant or planning a pregnancy. Line 100 states that there is sparse data available on social media usage in this age group regarding preconception/pregnancy information, but WHY is this data important to know? Are these women more vulnerable or at higher risk of poor health outcomes? Unplanned pregnancies?
  3. Line 81-84: these point are repeated in the discussion and could be removed from introduction.
  4. Overall the introduction is a bit long and repetitive and can be made more succinct.
  5. Why is age used as a predictor in the models since there is such a narrow age range of the population? The validity of the significant association between age and using social media for pregnancy related info is questionable given this narrow spread.
  6. Subheadings in Tables 1 and 3 could be moved to their own column as it is not very reader-friendly in current format.
  7. Authors should consider framing this study as an investigation of social media use for preconception information and pregnancy planning among young women given that nearly all the data obtained referred to preconception and future projections about pregnancy.

Author Response

(The authors gave the same response as above.)

Round 2

Reviewer 2 Report

The authors have adequately addressed my comments and improved the manuscript.